# Explainability of three-dimensional convolutional neural networks for functional magnetic resonance imaging of Alzheimer's disease classification based on gradient-weighted class activation mapping

**Boyue Song**[1]*, **Shinichi Yoshida**[2], **for the Alzheimer's Disease Neuroimaging Initiative**[¶]

**1** Graduate School of Engineering, Kochi University of Technology, Kami City, Kochi Prefecture, Japan,
**2** School of Information, Kochi University of Technology, Kami City, Kochi Prefecture, Japan

¶ Data used in preparation of this article were obtained from the Alzheimer's Disease Neuroimaging Initiative (ADNI) database (adni.loni.usc.edu). As such, the investigators within the ADNI contributed to the design and implementation of ADNI and/or provided data but did not participate in analysis or writing of this report. A complete listing of ADNI investigators can be found at: http://adni.loni.usc.edu/wpcontent/uploads/how_to_apply/ADNI_Acknowledgement_List.pdf

* 258005p@gs.kochi-tech.ac.jp

**Data Availability Statement:** All programming code and model files are available from following

## Abstract

Currently, numerous studies focus on employing fMRI-based deep neural networks to diagnose neurological disorders such as Alzheimer's Disease (AD), yet only a handful have provided results regarding explainability. We address this gap by applying several prevalent explainability methods such as gradient-weighted class activation mapping (Grad-CAM) to an fMRI-based 3D-VGG16 network for AD diagnosis to improve the model's explainability. The aim is to explore the specific Region of Interest (ROI) of brain the model primarily focuses on when making predictions, as well as whether there are differences in these ROIs between AD and normal controls (NCs). First, we utilized multiple resting-state functional activity maps including ALFF, fALFF, ReHo, and VMHC to reduce the complexity of fMRI data, which differed from many studies that utilized raw fMRI data. Compared to methods utilizing raw fMRI data, this manual feature extraction approach may potentially alleviate the model's burden. Subsequently, 3D-VGG16 were employed for AD classification, where the final fully connected layers were replaced with a Global Average Pooling (GAP) layer, aimed at mitigating overfitting while preserving spatial information within the feature maps. The model achieved a maximum of 96.4% accuracy on the test set. Finally, several 3D CAM methods were employed to interpret the models. In the explainability results of the models with relatively high accuracy, the highlighted ROIs were primarily located in the precuneus and the hippocampus for AD subjects, while the models focused on the entire brain for NC. This supports current research on ROIs involved in AD. We believe that explaining deep learning models would not only provide support for existing research on brain disorders, but also offer important referential recommendations for the study of currently unknown etiologies.

link: https://github.com/Neutrino000/3D-VGG-ADNI The authors do not own data used in the manuscript. Data obtained were collected and owned by the Alzheimer's Disease Neuroimaging Initiative. Researchers may request and access the data through the website of the Alzheimer's Disease Neuroimaging Initiative (ADNI) (http://adni.loni.usc.edu). Authors had no special access privileges to this data.

**Funding:** This work was supported by Japan Society for the Promotion of Science KAKENHI, Grant Numbers JP22K12786, JP22K19650, JP21H03553, JP22H03699, and JP20H00267, China Scholarship Council 202106030072. The funders had no role in study design, data collection and analysis, decision to publish, or preparation of the manuscript. Data collection and sharing for this project were funded by the Alzheimer's Disease Neuroimaging Initiative (ADNI) (National Institutes of Health Grant U01 AG024904) and DOD ADNI (Department of Defense award number W81XWH-12-2-0012). ADNI is funded by the National Institute on Aging, the National Institute of Biomedical Imaging and Bioengineering, and through generous contributions from the following: AbbVie, Alzheimer's Association; Alzheimer's Drug Discovery Foundation; Araclon Biotech; BioClinica, Inc.; Biogen; Bristol-Myers Squibb Company; CereSpir, Inc.; Cogstate; Eisai Inc.; Elan Pharmaceuticals, Inc.; Eli Lilly and Company; EuroImmun; F. Hoffmann-La Roche Ltd. and its affiliated company Genentech, Inc.; Fujirebio; GE Healthcare; IXICO Ltd.; Janssen Alzheimer Immunotherapy Research \& Development, LLC.; Johnson \& Johnson Pharmaceutical Research \& Development LLC.; Lumosity; Lundbeck; Merck \& Co., Inc.; Meso Scale Diagnostics, LLC.; NeuroRx Research; Neurotrack Technologies; Novartis Pharmaceuticals Corporation; Pfizer Inc.; Piramal Imaging; Servier; Takeda Pharmaceutical Company; and Transition Therapeutics. The Canadian Institutes of Health Research is providing funds to support ADNI clinical sites in Canada. Private sector contributions are facilitated by the Foundation for the National Institutes of Health (\href{blue}{www.fnih.org}). The grantee organization is the Northern California Institute for Research and Education, and the study is coordinated by the Alzheimer's Therapeutic Research Institute at the University of Southern California. The authors are thankful for the support.

**Competing interests:** The authors have declared that no competing interests exist.

## Introduction

Functional magnetic resonance imaging (fMRI) has been widely used for brain mapping research since the 1990s because of its ability to provide detailed functional information about the brain without requiring injections, surgery, or exposure to ionizing radiation [1]. Despite its significant advantages such as non-invasiveness, functional specificity, and high spatial resolution, fMRI is not commonly used for clinical diagnosis because of its susceptibility to noise and the complexity of its data [2, 3]. Since the useful signal variation in fMRI is generally only 2–5% of the signal strength, even a slight amount of noise can significantly affect the quality of data. Moreover, due to the relatively high temporal and spatial resolutions of fMRI, the complexity of its data is extremely high, which is another primary reason why fMRI data is hard to be directly utilized for medical analysis and diagnosis. Therefore, various statistical methods, such as independent components analysis (ICA) [4, 5] and general linear model (GLM) [6, 7], are often used to reduce data complexity and extract useful information from fMRI data. Tang et al. [8] proposed a method in which each brain was registered to MNI standard space and subdivided into 90 regions, and the regional time series were obtained by calculating the average fMRI signal across all voxels in each region. After that, functional connectivity between each pair of regions by calculating the Pearson correlation coefficient (PCC). With this method, features with good discriminative power were extracted and used to finish classification. In addition to general statistical methods, researchers have developed various dimensionality reduction techniques for fMRI data, such as the amplitude of low-frequency fluctuation (ALFF) [9], fractional ALFF (fALFF) [9], regional homogeneity (ReHo) [10], and voxel-mirrored homotopic connectivity (VMHC) [11]. With these methods, temporal information in the fMRI is compressed to generate only one brain volume, which significantly reduces the complexity of the data. In addition, different methods can be used to extract useful information from various perspectives.

However, traditional statistical methods require handcrafted feature extraction, which can result in inefficiencies and occasional errors. Recent advancements in computer hardware processing power and innovations in new graphical software have led researchers to focus increasingly on automation informed by deep learning. The combination of raw fMRI data and convolutional neural networks (CNNs) has recently enabled the automatic classification of various neurological diseases. As mentioned in [12], AD is the most common cause of dementia among older adults, a progressive disorder that starts with mild symptoms and worsens progressively. The progress from NC to AD can be subdivided into four stages, namely significant memory concern (SMC), early mild cognitive impairment (EMCI), mild cognitive impairment (MCI), late mild cognitive impairment (LMCI). Approximately 96.85% accuracy was achieved in classifying fMRI data from patients with AD compared to NC subjects [13]. The fMRI data were preprocessed using a standard pipeline, sliced into two-dimensional images from axial view and time axes, and randomly divided into training and testing datasets. LeNet-5 was used for classification. In a previous study [12], a similar pre-processing pipeline was employed; however, residual neural networks [14] were utilized to classify AD. By fine-tuning transfer learning, an accuracy of 97.88% was achieved. A modified three dimensional (3D) CNN has been applied in resting-state fMRI data [2], in which four-dimensional fMRI data were sliced along the time axis to aid training. This method was approximately 98.96% accurate for AD. [15] developed a robust low-cost neural network classification system for AD and Mild Cognitive Impairment (MCI) against NC using a CNN with input images based on diffusion maps and gray-matter volumes, achieving competitive results of 93.5% for AD/NC classification. In another study relating raw MRI data [16], the authors employed ResNet-50 and LeNet on AD classification based on MRI slices in three views and categories. The study

demonstrated that the selecting slices performed better than using entire slices in MRI images for AD classification and the coronal view showed higher accuracy. Although these methods demonstrate high accuracy, their models lack explainability, thereby diminishing the credibility of the model due to the inability to explain its predictions.

Deep learning models extract features from the input data based on labeled data distributions and then make predictions based on these features. The use of raw fMRI data would evidently increase the difficulty in extracting useful information for the model, which would subsequently affect performance. In some studies, statistical methods or other means have been employed to reduce the complexity of raw fMRI data as a preprocessing step to reduce the difficulty of feature extraction by the model. A framework for the early diagnosis of AD has been developed using deep neural networks and various medical information, in which functional brain networks were constructed from resting-state fMRI signal correlations and used as correlation coefficient data to train the neural network [17]. Similarly, [18] proposed a deep learning-based method to realize binary classification between each pair of the different stages of AD. The accuracy exceeded 99%. In addition, [19] reduced the dimensionality of fMRI data by extracting features as 3D spatial maps for classifying resting-state fMRI images using a 3D-CNN. An accuracy of 85.27% was obtained for the binary classification of AD versus NC. Dimensionality reduction method have not only been applied to AD, but also to schizophrenia. In [20], Group ICA was considered as a preprocessing step for extracting ICA components from a schizophrenia dataset, and 3D-CNN was employed to complete the classification. Furthermore, 98.09% ten-fold cross-validated classification accuracy was achieved. In [3], fMRI images were preprocessed, and functional connectivity analysis was used to extract features. Subsequently, 3D-CNN and a long short-term memory recurrent network were utilized to extract spatial and temporal information for classifying functional activity maps. They achieved an accuracy of 92.32% for the Center for Biomedical Research Excellence dataset [21]. [22] investigates the utility of correlated transfer function (CorrTF) as a novel biomarker for extracting crucial features from resting-state fMRI data. Employing a support vector machine (SVM) in hierarchical and flat multi-classification schemes, the research achieved competitive results of 98.2% for distinguishing between various stages of AD. In our opinion, the more detailed the manually extracted features are in the entire classification task, the simpler the process of automatic feature extraction required by the deep learning models; thus, the performance of the models may be better. However, we cannot guarantee that the manually extracted features are precisely the features required by the model for classification. Therefore, there is a trade-off between manually pre-extracting features and allowing the model to automatically extract features.

Although deep learning has made significant achievements in various fields, it is frequently referred to as a black box because of its lack of explainability, which means that the underlying reasons for a given prediction cannot be ascertained. This holds true whether the prediction is accurate or not and can greatly impact the reliability of the model, particularly in clinical diagnosis. A method called class activation mapping (CAM) was proposed in 2016 to visualize the model [23]. In this approach, the final fully connected layer is replaced by a global average pooling (GAP) layer and feature maps from the last convolutional layer are used to visualize the model. A novel method called Gradient-weighted CAM (Grad-CAM), which builds upon CAM by combining the gradients of the gradient descent algorithm with the feature maps from the final convolutional layer was introduced in 2017 [24]. Subsequently, multiple CAM-based methods have been proposed [25–31], and the explainability of models has become an increasingly important direction in computer vision.

In recent years, visualization techniques have been employed to explain deep learning models based on MRI images. In 2019, [32] proposed using layer-wise relevance propagation

(LRP) to visualize CNN decisions for AD based on structural MRI (sMRI) data. The results showed that a lot of importance is put on areas in the temporal lobe including the hippocampus. [33] also proposed a CNN for the detection of AD based on sMRI. In this work, the association of relevance scores and hippocampus volume were evaluated to validate the clinical utility. A high accuracy ($AUC \geq 0.91$) was achieved for AD versus NC. Relevance maps indicated that hippocampal atrophy was found the most informative factor for AD detection. [34] proposed a 3D-CNN framework using a spatial source phase (SSP) maps derived from complex-valued fMRI data to classify schizophrenia patients (SZ) and NC. Grad-CAM was utilized to localized all contributing ROIs with opposite stengths for SZ and NC. [35] employed CNN trained on three orthogonal views of cerebral regions, specifically the hippocampi, amygdalae, and insulae, to stage the AD spectrum, including preclinical AD, MCI, AD, and NC, using patched from structured MRI. The performance is comparable to state-of-art methods, showcasing the potential of patch-based region of interest (ROI) ensembles in providing informative landmarks for MRI analysis. In addition to MRI images, [36] presented a deep learning system designed to automatically identify four visually explainable signs of emphysema in frontal and lateral chest radiographs, providing explainable labels for the detected signs. [37] leverages a neural network model trained on synthetic NaI(Tl) urban search data to assess and adapt explanation methods for gamma-ray spectral data. It highlights the superior accuracy of black box methods, specifically LIME and SHAP, with a preference for SHAP due to its minimal hyperparameter tuning.

Extensive research have explored the application of fMRI based deep neural networks to diagnose neurological disorders. Despite this, only a limited number of studies have provided results regarding explainability. In this study, we applied several CAM methods to two 3D-VGG16 models, which were used to classify patients with AD and NC based on four types of 3D resting-state functional activity maps. AD is a neurological disorder characterized by the degeneration of memory-related neurons in the brain. As the disease progresses, different regions of the brain exhibit varying patterns of blood oxygenation levels [38, 39]. Blood oxygenation level refers to the proportion of oxygen bound hemoglobin, which can be used to infer brain activity, as neural activity induces alterations in local blood flow and oxygenation levels [40]. We hypothesized that fMRI captures blood oxygenation patterns in AD-affected ROIs. Deep learning models classify AD stages using these patterns and generate Grad-CAM heatmaps to identify the affected ROIs. The use of heatmaps to explain the model's focus on specific ROIs can not only assist and support researchers in studying diseases with known and unknown causes. Besides, analyzing the heatmaps of cases, where the model made prediction errors, can help improve the performance of the model. The main aims of our work are as follows:

- To utilize several resting-state functional activity maps as dataset instead of raw fMRI data, which can manually assist the model in extracting pertinent information

- To replace fully connected layers with GAP layer in the model serves to preserve spatial information within feature maps, mitigate overfitting, and enhance model performance

- To employ 3D CAM methods on an fMRI-based 3D-VGG16 model for AD diagnosis, validating the model's efficacy, and identifying specific ROIs as the basis for classification, potentially indicating AD lesions

First, we introduce the dataset used, as well as the preprocessing steps, deep learning frameworks, and Grad-CAM method in the Materials and methods section. In the Results section, we present the experimental results, including the model's performance and 3D Grad-CAM heatmaps of 3D-VGG16 networks. In the Discussion section, the performance of the models

and the Grad-CAM results are discussed. The final section concludes with a summary of the main findings and contributions of the study.

## Materials and methods

### Alzheimer's Disease Neuroimaging Initiative (ADNI) dataset and preprocessing

Here, we introduce the dataset used in our study, as well as the preprocessing steps, deep learning frameworks. The program is publicly available at https://github.com/Neutrino000/3D-VGG-ADNI. Data used in the preparation of this article were obtained from the Alzheimer's Disease Neuroimaging Initiative (ADNI) database (adni.loni.usc.edu). The ADNI was launched in 2003 as a public-private partnership, led by Principal Investigator Michael W. Weiner, MD. The primary goal of ADNI has been to test whether serial magnetic resonance imaging (MRI), positron emission tomography (PET), other biological markers, and clinical and neuropsychological assessment can be combined to measure the progression of mild cognitive impairment (MCI) and early Alzheimer's disease (AD). All ADNI studies are conducted according to the Good Clinical Practice guidelines, the Declaration of Helsinki, and U.S. 21 CFR Part 50 (Protection of Human Subjects) and Part 56 (Institutional Review Boards). Written informed consent was obtained from all participants before protocol-specific procedures were performed. The Institutional Review Boards approved the ADNI protocol of all participating institutions; for up-to-date information, see www.adni-info.org. Based on multiple scans obtained at various time points for each subject, the dataset for our study consisted of 163 scans of fMRI data from 50 NCs and 105 scans of fMRI data from 34 patients with AD, which implies that some subjects possess multiple sets of scan data. Table 1 presents some characteristics of the ADNI dataset. The small size of datasets is a common issue in medical data. Therefore, the typical practice is to include all samples [19, 41–43], which may result in data imbalance. However, larger datasets often offer better generalization performance, leading to a trade-off. A standardized preprocessing pipeline is employed to process the ADNI dataset. The pipeline included various steps to remove noise and improve generality.

First, the dataset was converted from the Digital Imaging and Communications in Medicine (DICOM) format to Neuroimaging Informatics Technology Initiative (NIFTI) format using the dcm2niix toolbox [44]. Subsequently, Data Processing and Analysis for Brain Imaging (DPABI) [45] on the MATLAB 9.12.0 (2022a) platform were applied to the remaining preprocessing steps. Brain extraction was performed using anatomical and functional images. Subsequently, temporal adjustment was achieved through slice-timing correction, while the influence of head motion on data acquisition was removed by motion correction. In addition, the entire dataset was subjected to intensity normalization to ensure that the mean intensity remained consistent and uniform. Spatial registration was then conducted to align the fMRI images from the participants' individual spaces to the standard space of the MNI152 template. Finally, a 4-mm full-width at half-maximum (FWHM) cubic Gaussian filter was used for

**Table 1. Characteristics of the ADNI dataset.**

| Study | Number of subjects | fMRI scans | Mean Age |
|---|---|---|---|
| NC | 50 | 163 | 74.80 |
| AD | 34 | 105 | 74.68 |
| Total | 84 | 268 | |

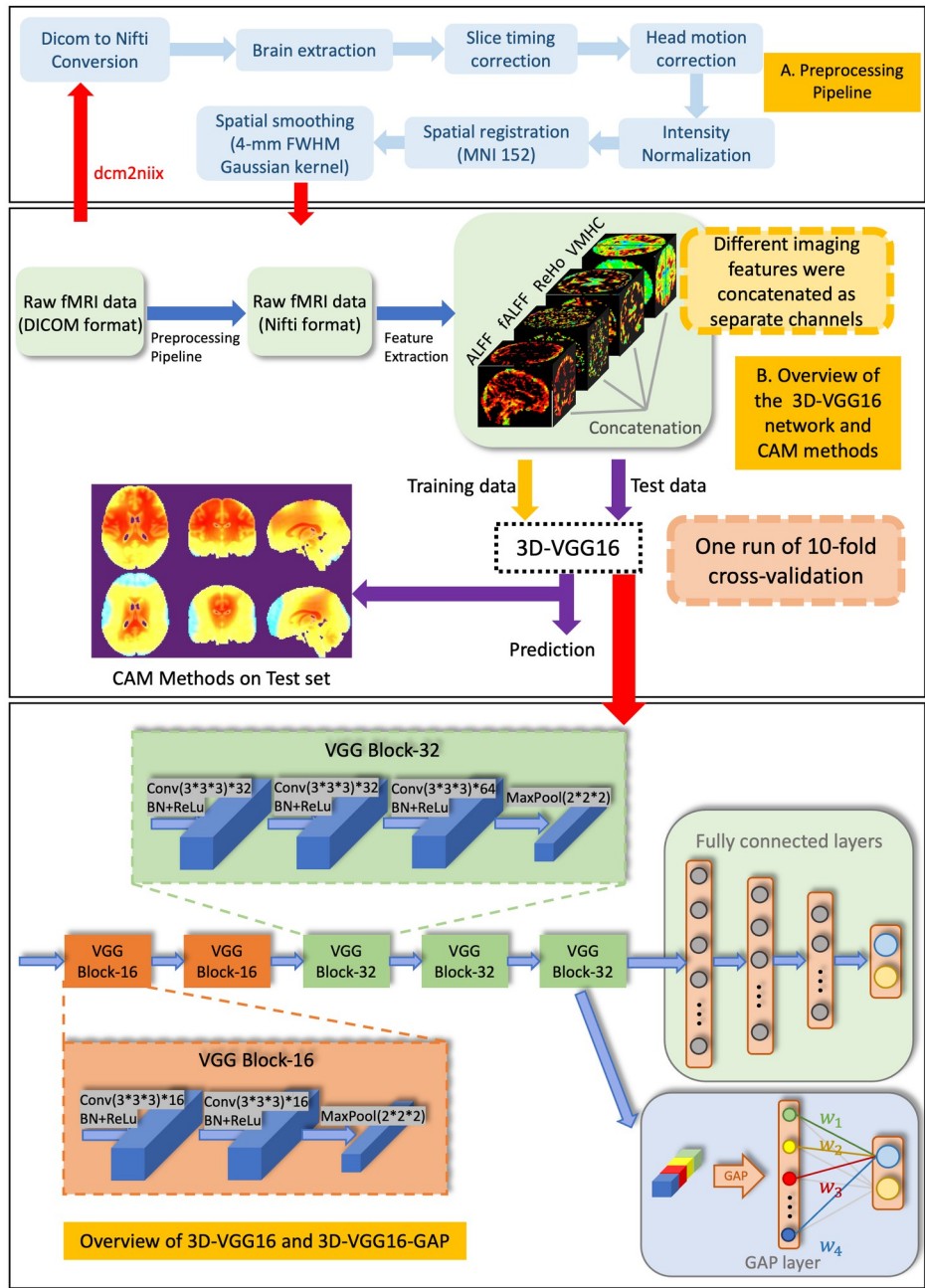

**Fig 1. Overview of the proposed 3D-VGG16-GAP in AD classification.** A. The standard preprocessing pipeline used in this study. B. Overview of the proposed 3D-VGG16 network and Grad-CAM methods. C. Framework of 3D-VGG16 and 3D-VGG16-GAP.

spatial smoothing during the application of deconvolution to the 3D images. The standard pre-processing pipeline is shown in Fig 1A.

After all the preprocessing steps was completed, a data matrix of sized $61 \times 73 \times 61 \times 140$ was obtained for each participant, where each volume consisted of $61 \times 73 \times 61$ data points recorded over 140 time points.

## Resting-state functional activity maps

fMRI is an imaging technique that can capture information with high spatial resolution and relatively high temporal resolution. It can detect activity in the ROIs within a spatial range of a few millimeters with a temporal resolution of several seconds to tens of seconds. However, the high-dimensional images indicate the high complexity of the data, which limits accurate analysis and description. Although deep learning excels at extracting information from massive amounts of data, highly complex data can still affect the performance of the model.

In this study, four different resting-state functional activity maps-ALFF [9], fALFF [9], ReHo [10], and VMHC [11, 46, 47] were extracted from resting-state fMRI data which describe fMRI data from different aspects but with lower complexity [3]. All resting-state functional activity maps were obtained using DPABI. Fig 2 showed these four kinds of resting-state functional activity maps between NC and AD subjects.

**ALFF.** After preprocessing, the fMRI data were temporally band-pass filtered ($0.01 < f < 0.1$Hz) to eliminate low-frequency noise from drift and high-frequency noise from respiratory and cardiac activity. Using the fast Fourier transform (FFT), the time series of each voxel were transformed into the frequency domain:

$$x(t) = \sum_{k=1}^{N}[a_k\cos(2\pi f_k t) + b_k\sin(2\pi f_k t)] \tag{1}$$

Since the power of a specific frequency is proportional to the square of the corresponding amplitude, the mean square root of the power spectrum across a frequency range of $0.01 - 0.1$Hz for each voxel was computed as follows:

$$\text{ALFF} = \sum_{K:f_k\in[0.01,0.1]}\sqrt{\frac{a_k^2(f) + b_k^2(f)}{N}} \tag{2}$$

**fALFF.** fALFF is a variant form of ALFF, which can further reduce the physiological noise by considering the ratio of each frequency ($0.01 < f < 0.1Hz$) to the total frequency range. In addition, the application of fALFF enhances both the sensitivity and specificity of detecting

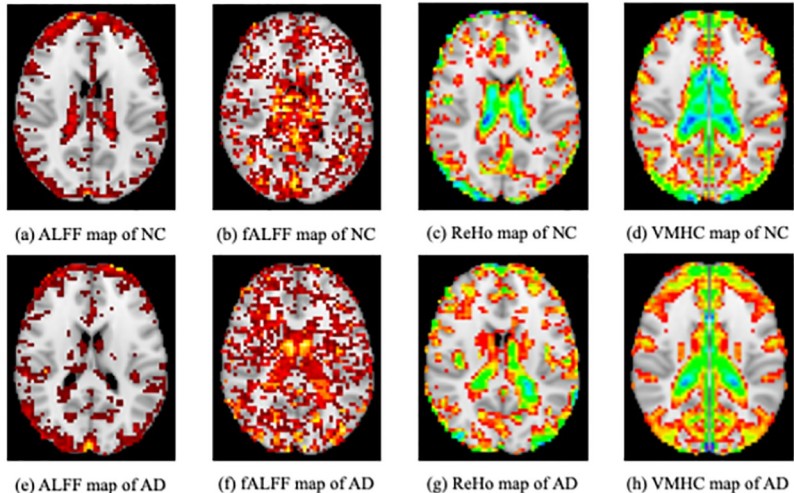

(a) ALFF map of NC  (b) fALFF map of NC  (c) ReHo map of NC  (d) VMHC map of NC

(e) ALFF map of AD  (f) fALFF map of AD  (g) ReHo map of AD  (h) VMHC map of AD

**Fig 2. Resting-state functional activity maps.**

spontaneous activity. fALFF is calculated as follows:

$$\text{fALFF} = \frac{\sum_{K:f_k \in [0.01,0.1]} \sqrt{\frac{a_k^2(f) + b_k^2(f)}{N}}}{\sum_{K=1}^{N} \sqrt{\frac{a_k^2(f) + b_k^2(f)}{N}}} \tag{3}$$

**ReHo.** ReHo measures the similarity of the time series of a given voxel to those of its nearest neighbors in a voxel-wise manner using Kendall's coefficient concordance (KCC) [48]. ReHo is calculated as follows:

$$\text{KCC} = \frac{\sum R_i^2 - n(\bar{R}^2)}{\frac{1}{12} k^2 (n^3 - n)} \tag{4}$$

in which $R_i$ is the number rank of the $i$th time point; $\bar{R} = ((n+1)K)/2$ is the mean of $R_i$; $K$ is the number of time series within a measured cluster ($K = 27$ in our study, which means one given voxel plus the number of its neighbors); and $n$ is the number of ranks.

**VMHC.** VMHC is a method employed in the analysis of fMRI scans that enables the assessment of functional similarity between the two hemispheres of the brain. To generate VMHC maps, the fMRI data of each participant were used to compute the PCC between a particular voxel and its corresponding voxel in the opposite hemisphere, followed by the application of Fisher's z-transform to enhance the normality of the values. It is calculated as follows:

$$r_{x,y} = \frac{\sum_{i=1}^{n} (X_i - \bar{X})(Y_i - \bar{Y})}{\sqrt{\sum_{i=1}^{n} (X_i - \bar{X})^2} \sqrt{\sum_{i=1}^{n} (Y_i - \bar{Y})^2}} \tag{5}$$

## Deep learning framework

After obtaining all the resting-state functional activity maps, 3D VGG was used to finish feature extraction and classification.

**Three-dimensional VGG.** VGG16 [49] is a deep CNN model developed by the Visual Geometry Group. It is part of the VGG family of models and was designed for image classification tasks. In this study, a 3D version of the VGG16 model was used, as illustrated in Fig 1B and 1C.

The 3D-VGG16 model consisted of 16 layers, including 13 3D convolutional layers and three fully connected layers. The entire model was divided into five VGG blocks, with the first two VGG blocks containing two convolutional layers each, and the last three VGG blocks containing three convolutional layers each. There was a max-pooling layer with a $2 \times 2 \times 2$ filter and stride of $2 \times 2 \times 2$ voxels at the end of each VGG block. The convolutional layers were composed of $3 \times 3 \times 3$ filters with a stride of one voxel and padding of one voxel, each of which was followed by a batch normalization layer and rectified linear unit (ReLU) as the activation layer. The fully connected layers had 2, 048 and 1, 024 units separately, and the final layer was a softmax layer with two units corresponding to the two AD and NC classes.

**Global average pooling layer.** Since the CAM method [23] must expectedly be used for the explainability of 3D-VGG16 in our study, we replaced the fully connected layers at the end of the model with a global average pooling layer (GAP) layer [50], which is a necessary structure for CAM method. It is worth noting that GAP layer is not required for other CAM methods, such as Grad-CAM. The model with the GAP layer can retain its remarkable localization

ability until the final layer and easily identify discriminative image regions. In addition, GAP layer can also be employed to prevent overfitting due to the reduced number of parameters [51]. As shown in Fig 1C, the GAP layer computes the average value of each feature map unit in the last convolutional layer, which is then combined using a weighted sum to produce the final output of the model. In our experiment, we employed five CAM methods, including the original CAM [23], Grad-CAM [24], Grad-CAM plus plus [25], Eigen-CAM [26] and Eigen Grad-CAM. However, there was little difference between the heatmaps generated using the different CAM methods. Therefore, owing to space limitations, only the results obtained using the Grad-CAM method are presented in the Results section.

## Grad-CAM

Grad-CAM [24] is an improved method based on CAM. The gradient information that flowed into the last convolutional layer of the model was used to determine the significance of each neuron in making predictions. In deep learning models, the features extracted by convolutional layers become increasingly high-level as they progress deeper into the network. Therefore, we chose to utilize the feature maps corresponding to the last convolutional layer. It is worth noting that Grad-CAM can be applied beyond the last convolutional layer in the neural network architecture, allowing for its utilization across multiple layers for visualizing the importance of features.

First, the gradients of the score for a particular class, $c$, which was denoted by, $y^c$, was computed with respect to the feature maps, $A^k$, of the last convolutional layer, $i.e.$ $\partial y^c / \partial A^k$ in the last convolutional layer. Subsequently, global average pooling was employed on the gradients to calculate the neuron importance weights, $\alpha_k^c$, as follows:

$$\alpha_k^c = \frac{1}{Z} \sum_i \sum_j \frac{\partial y^c}{\partial A_{ij}^k} \tag{6}$$

The weight, $\alpha_k^c$, represents the importance of the feature map, $k$, in predicting the target class, $c$. $Z$ signifies the number of elements in $A^k$.

Subsequently, a weighted combination of forward feature maps was applied. Finally, ReLU was employed to remove the negative values as follows:

$$L_{Grad-CAM}^c = \text{ReLu}\left( \sum_k \alpha_k^c A^k \right) \tag{7}$$

A coarse heat-map was generated with the same resolutions as the feature maps of the final convolutional layer. Once the coarse Grad-CAM heat map was obtained, bilinear interpolation algorithms are required to match the resolutions of the original images, which enables the visualization and comparison of the results in a more intuitive manner.

## Results

### Experimental setup

The models were built and trained using Python 3.6 with pytorch 1.9.0. on a Linux machine with 512 GB RAM and 32 GB NVIDIA GPU card. To compare the discriminative power, each functional activity map was trained as a dataset individually, aside from combining maps trained as the dataset. For the combined maps, we combine each individual functional activity maps together in a manner akin to assembling RGB channels in natural images.

The Adam optimizer that has a learning rate of $5 \times 10^{-5}$ was utilized. Due to the small size of ADNI dataset, a small value of 4 is set as batch size, which could enhance the model's

generalization by introducing more randomness in each batch to help prevent overfitting and improve model's performance on unseen data. Furthermore, due to the limited exposure to data in each batch, the model is compelled to acquire more generalized features rather than memorizing specific samples from the entire dataset. The decay rate of the weights was set to $5 \times 10^{-4}$. Given that various train/test splits result in dramatically different rankings of models [52, 53], each model was trained 10 times to obtain the average accuracy for the generality and robustness of the model. For every training session, 80% of the data was randomly selected as the training set, whereas 20% was randomly selected as the testing set for prediction. It is noteworthy that, given the presence of multiple scans for certain subjects, subject-level data split is employed to prevent data leakage. Furthermore, each scan corresponds to only one set of resting state functional activity maps. Finally, for normal 3D-VGG16 without a GAP layer, a dropout layer was applied in each fully connected layer at a rate of 0.5.

## Evaluation of deep learning models

As shown in Tables 2 and 3, the accuracies of the combined maps were 91.1% and 96.4% for 3D-VGG16 and 3D-VGG16-GAP, respectively. When a single functional activity map was used as dataset, the ReHo map achieved the highest accuracy of 87.5% on 3D-VGG16, whereas the ALFF map and ReHo maps achieved the highest accuracy of 91.1% on 3D-VGG16-GAP. Combining maps achieved the highest average accuracies of 84.1% and 87.9% for both two models, respectively, which may be due to the most comprehensive information contained in the dataset. For 3D-VGG16, both the ALFF and ReHo maps achieved an average accuracy of approximately 80%, the VMHC map obtained an accuracy of approximately 72%, and the fALFF map had the lowest accuracy at 66.8%. For 3D-VGG16-GAP, the average accuracy of the ALFF map at 84.7% was second only to that of the combined maps. The ReHo map achieved an accuracy of approximately 82%, which was 7% higher than that of the VMHC map. The accuracy of the fALFF map was the lowest among the functional activity maps.

Overall, 3D-VGG16-GAP performed better than 3D-VGG16, which may be due to the effective reduction in the parameters and suppression of overfitting by replacing the fully connected layers with the GAP layer [52]. In addition, the combined maps achieved the highest

**Table 2. Accuracy of 3D-VGG16 with ten training runs.**

|  |  | 1 | 2 | 3 | 4 | 5 | 6 | 7 | 8 | 9 | 10 | Average |
|---|---|---|---|---|---|---|---|---|---|---|---|---|
| (%) | ALFF | **83.9** | 83.9 | 76.8 | 76.8 | 83.9 | 80.4 | 78.6 | 82.1 | 80.4 | 80.4 | 80.7 ± 2.6 |
|  | fALFF | 69.6 | 60.7 | 66.1 | 64.3 | 64.3 | 67.9 | 64.3 | 71.4 | 64.3 | **75.0** | 66.8 ± 4.0 |
|  | ReHo | 69.6 | **87.5** | 87.5 | 82.1 | 75.0 | 78.6 | 82.1 | 87.5 | 71.4 | 78.6 | 80.0 ± 6.2 |
|  | VMHC | 76.8 | 78.6 | 62.5 | **80.4** | 76.8 | 67.9 | 67.9 | 66.1 | 62.5 | 78.6 | 71.8 ± 6.7 |
|  | Combined | 83.9 | 83.9 | **91.1** | 87.5 | 82.1 | 82.1 | 82.1 | 82.1 | 80.4 | 85.7 | **84.1 ± 3.0** |

**Table 3. Accuracy of 3D-VGG16-GAP with ten training runs.**

|  |  | 1 | 2 | 3 | 4 | 5 | 6 | 7 | 8 | 9 | 10 | Average |
|---|---|---|---|---|---|---|---|---|---|---|---|---|
| (%) | ALFF | 80.4 | **91.1** | 80.4 | 83.9 | 78.6 | 80.4 | 89.3 | 89.3 | 87.5 | 85.7 | 84.7 ± 4.3 |
|  | fALFF | 73.2 | 78.6 | 71.4 | 69.6 | 71.4 | 64.3 | **80.4** | 73.2 | 67.9 | 75.0 | 72.5 ± 4.5 |
|  | ReHo | 83.9 | 78.6 | 80.4 | 85.7 | 78.6 | 78.6 | 75.0 | 85.7 | **91.1** | 80.4 | 81.8 ± 4.5 |
|  | VMHC | 75.0 | 69.6 | 82.1 | 75.0 | 75.0 | 66.1 | **85.7** | 73.2 | 76.8 | 73.2 | 75.2 ± 5.3 |
|  | Combined | 87.5 | 85.7 | 85.7 | 83.9 | 89.3 | 85.7 | 85.7 | 91.1 | 87.5 | **96.4** | **87.9 ± 3.5** |

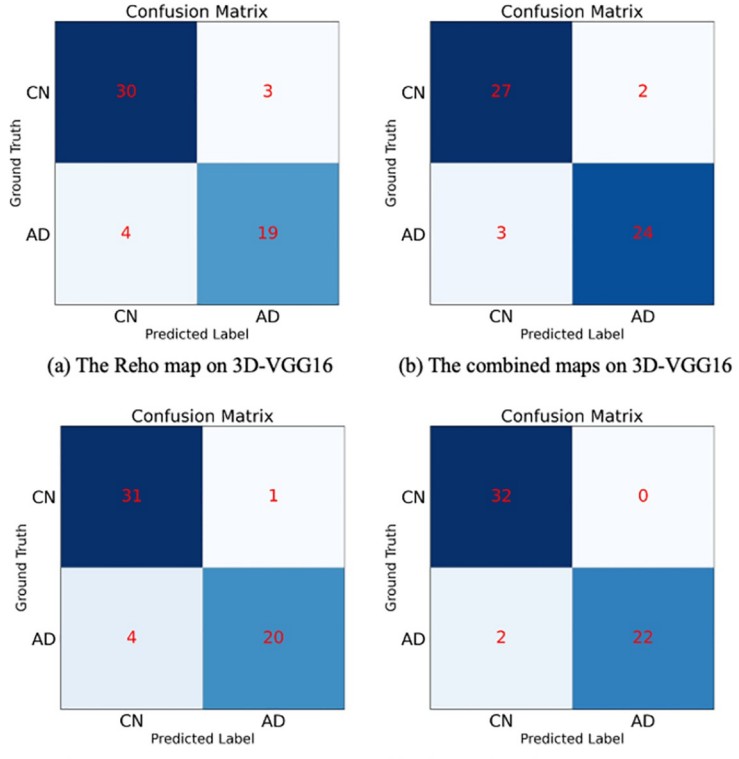

**Fig 3. Confusion Matrix of AD and NC for ReHo map on 3D-VGG16 (a), combined maps on 3D-VGG16 (b), ALFF map on 3D-VGG16-GAP (c), and combined maps on 3D-VGG16-GAP (d).**

accuracy for both models. For the single functional activity map, the performances of the ALFF and ReHo maps were better than that of the VMHC map, whereas the fALFF map had the lowest accuracy. The confusion matrices of the different maps of the two models are shown in Fig 3.

## Explainability of deep learning models based on Grad-CAM

In this section, 3D version of the Grad-CAM heatmaps of the two models are presented. It must be noted that in our experiment, we employed five CAM methods, including the original CAM [23], Grad-CAM [24], Grad-CAM plus plus [25], Eigen-CAM [26] and Eigen Grad-CAM. However, there was little difference between the heatmaps generated using the different CAM methods. Therefore, owing to space limitations, only the results obtained using the Grad-CAM method are presented.

Similar to some studies that proposed CAM methods [23–26], we used the heatmaps shown in Figs 4 and 5 to visually demonstrate the results of Grad-CAM. Because the coarse heat map has the same resolution as the feature map of the last convolutional layer, which is only 14 × 14 × 14, it is difficult to compare it visually with the original image. Therefore, bilinear interpolation algorithms were employed to match the resolutions of the Grad-CAM and original brain images.

Figs 4 and 5 show the average Grad-CAM images of all the test samples of the two models, respectively. As shown, the left column shows the orthographic projection of the Grad-CAM images for the NC category based on different resting-state functional activity maps, while the

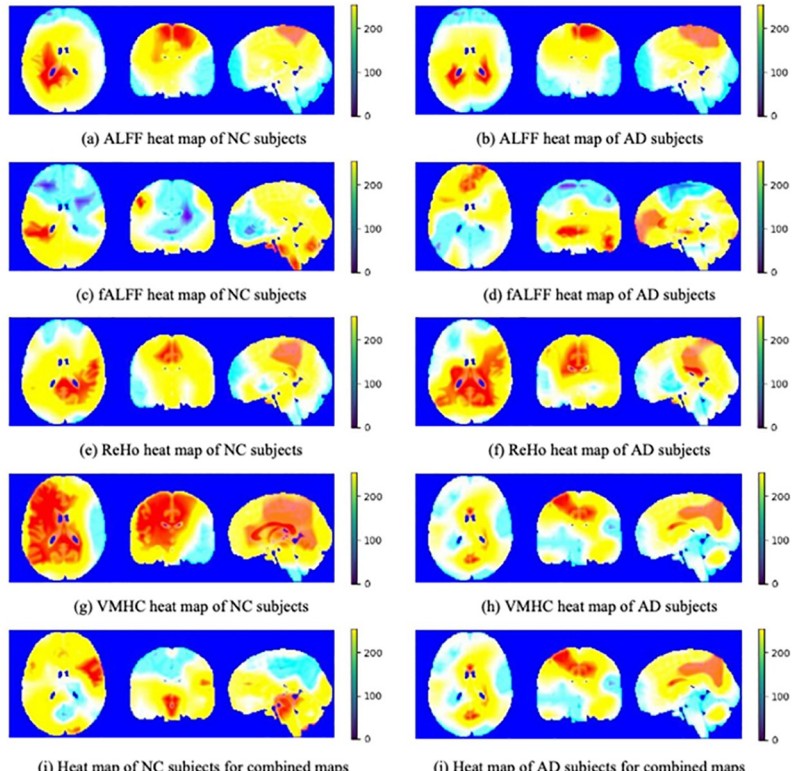

**Fig 4. Average heatmaps based on Grad-CAM method of 3D-VGG16.** The Grad-CAM heatmaps of the ALFF map for NC (a) and AD (b), the ReHo map for NC (e) and AD (f), and the combined maps for NC (i) and AD (j) tend to highlight some specific ROIs such as precuneus and hippocampus, corresponding to the fact that the accuracy of the ALFF map, the ReHo map, and the combined maps is relatively high. However, the heatmaps of the fALFF map for NC (c) and AD (d), and the VMHC map for NC (g) and AD (h) cannot locate any specific regions, which could be the reason for the lower accuracy of both. In addition, due to the need to flatten the final feature maps into vector form in 3D-VGG16, the spatial structure of the feature maps is destroyed, resulting in the poor imaging performance.

right column shows the orthographic projection of the Grad-CAM images for the AD category. In the NC category, the model tended to focus on the entire brain without focusing on any specific local region. However, for AD, the model tended to focus more on specific local regions. Because of the low initial resolution of Grad-CAM images (*i.e.* $14 \times 14 \times 14$), even after interpolation, they can only provide rough localization of a certain region and cannot achieve precise localization of a specific brain area. Compared with that, the precuneus region was highlighted in all Grad-CAM images, as shown in the left column of Fig 6. In addition, exception for fALFF, which had the lowest accuracy, the heatmaps of all other functional activity maps highlighted the hippocampal region, as shown in the right column of Fig 6.

The Grad-CAM heatmaps of the two patients that were incorrectly predicted are shown in Fig 7. The model focused on areas other than the precuneus and hippocampus, resulting in inaccurate feature extraction and ultimate incorrect prediction.

## Discussion

In this section, we explore the potential of deep learning techniques for distinguishing AD from NC using fMRI data. As mentioned in the Introduction, large quantities of information can be extracted from fMRI data owing to its high temporal and spatial resolutions. Generally,

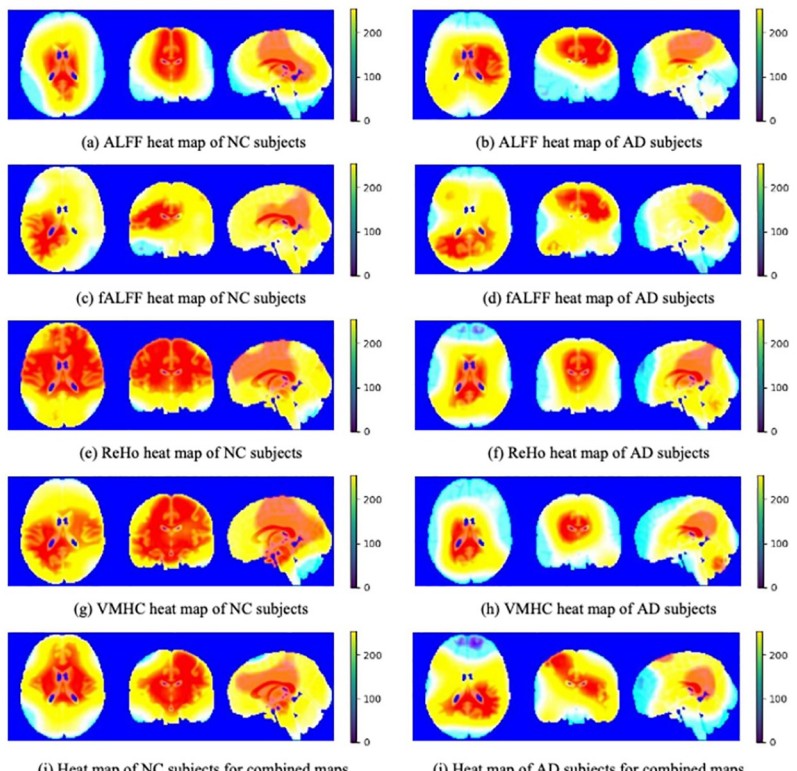

**Fig 5. Average heatmaps based on Grad-CAM method of 3D-VGG16-GAP.** Unlike 3D-VGG16, fully connected layers are replaced with GAP layer in 3D-VGG16-GAP, so that the spatial structure of the final feature maps is retained, resulting in the better imaging performance. All the Grad-CAM heatmaps for NC subjects (ALFF (a), fALFF (c), ReHo (e), VMHC (g), and combined maps (i)) tend to focus the whole brain, whereas the the Grad-CAM heatmaps for AD subjects (ALFF (b), fALFF (d), ReHo (f), VMHC (h), and combined maps (j)) tend to highlight some specific ROIs such as precuneus and hippocampus.

the relevant signal variation accounts for only 2% to 5% of the overall signal strength. Consequently, even a minor amount of noise can significantly affect the data. Additionally, a collection of fMRI data can consist of millions of data points because each voxel is scanned in both space and time, resulting in a relatively high level of complexity in the data. In our study, four different kinds of resting-state functional activity maps were used to extract the useful data from four different aspects.

As shown in Tables 2 and 3, our findings demonstrate that utilizing functional activity maps can result in relatively high accuracy in diagnosing AD from the ADNI dataset. Overall, the classification performance of 3D-VGG16-GAP was superior to that of 3D-VGG16. However, [54] reported a prediction performance loss because of the GAP layer, in which sMRI of ADNI serves as dataset, and 3D-VGG and 3D-ResNet are employed for classification. We believe that the main rationale of the performance loss overturned as a benefit is the difference of dataset. In our method, resting-state fMRI is utilized as the raw data from which several resting-state functional activity maps are extracted to form the dataset. fMRI and sMRI images capture different types of information. While fMRI reflects dynamic changes in brain and may benefit from GAP in capturing global properties due to stronger inter-regional correlations, sMRI depicts the anatomical structure of the brain, with lower inter-regional correlations, potentially leading to information loss with GAP application, resulting in performance loss.

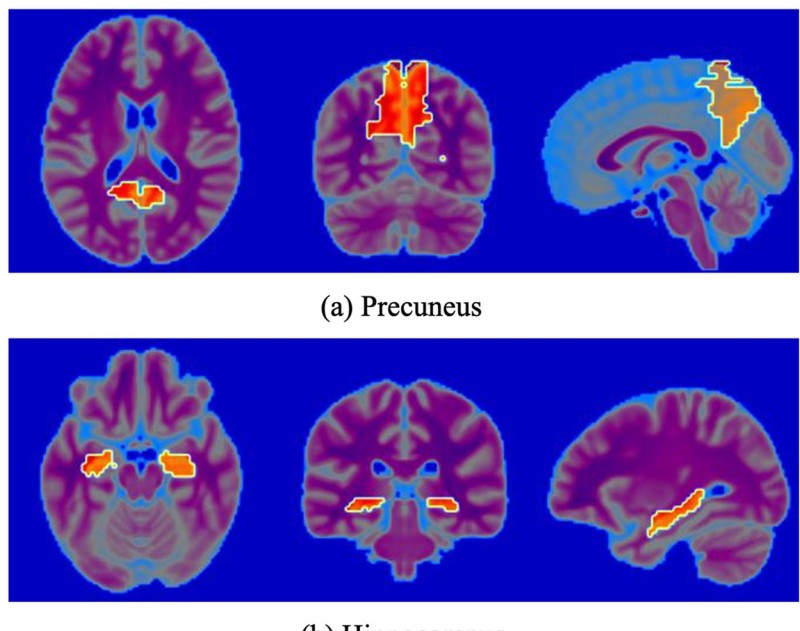

(a) Precuneus

(b) Hippocampus

**Fig 6. ROIs highlighted by deep learning models.** As shown in Fig 5, all the Grad-CAM heatmaps cover the precuneus. In some cases, such as the heatmaps of the ReHo, VMHC, and combined maps, the hippocampus is also covered.

The use of the GAP layer also reduces the number of model parameters and effectively mitigating overfitting, which is very common in medical image classification tasks with limited samples.

By comparing the results obtained using a single functional activity map versus combining maps, we found that the classification accuracy increased when the combined maps were applied as dataset. For the single functional activity map, the ALFF map exhibited an accuracy value second only to the combined maps on both models because of its precise reflection of the intensity of neuronal activity. Except for these two methods, the ReHo map achieved relatively high accuracies of 80.0% and 81.8% for 3D-VGG16 and 3D-VGG16-GAP, respectively. The ReHo map describes the local functional connectivity of a voxel to neighboring voxels, which

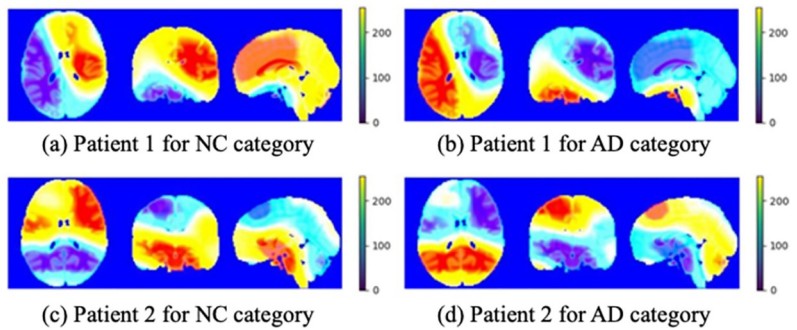

(a) Patient 1 for NC category            (b) Patient 1 for AD category

(c) Patient 2 for NC category            (d) Patient 2 for AD category

**Fig 7. Grad-CAM heatmaps of samples that were predicted incorrectly on 3D-VGG16-GAP.**

**Table 4. Comparative analysis with previous research on resting-state fMRI.**

| Study | Accuracy | Participants (n) | Data Source | Methods |
|---|---|---|---|---|
| Duc et al. [40] | 85.27% | 331 | Private | 3D-CNN |
| Gupta et al. [41] | 81% | 88 | ADNI | FNN |
| Lu et al. [43] | 71.9% | 60 | ADNI | Autoencoder |
| Qiao et al. [55] | 95.59% | 68 | Private | 2D-DAGNN |
| Bi et al. [56] | 94.44% | 60 | ADNI | RSVM-C |
| Proposed Method | 87.9% | 84 | ADNI | 3D-VGG16-GAP |

may be an important indicator for diagnosing AD becasue of the strong separation performance of the model. The accuracy of the VMHC map was lower than that of the combined, ALFF, and ReHo maps. The VMHC map measures functional homotopic connectivity between a voxel and its mirrored voxel in the contralateral brain hemisphere. The reason for the low accuracy of the VMHC map may be because there was no strong difference in functional homotopic connectivity between the AD and NC categories. Additionally, the models of VMHC map demonstrate the highest variance across all the maps, indicating its pronounced instability in performance on random datasets. The key distinction of VMHC map from other maps lies in its computation of the functional homotopic connectivity between the left and right hemispheres, resulting in symmetric data. We hypothesize that this symmetry may contribute to the model's susceptibility to overfitting on random split datasets, leading to unstable performance. Finally, the accuracy of the fALFF map was the lowest. The difference between the fALFF and ALFF maps was that the effect of noise is reduced and suppressed by considering the ratio of each frequency to the total frequency range. However, we believe that, while the influence of noise is reduced, the intensity of some useful information required for model classification, such as neuronal activity signals, may also be suppressed, which may be the reason for the poor performance of the model. Table 4 presents the performance comparison between the proposed method and previous studies, demonstrating that our method is competitive and promising.

In general, CNNs extract features from input images in a hierarchical manner. Therefore, the convolutional layers closer to the front of the model extract the lower-level features, whereas those closer to the back extract the higher-level features. The final fully connected layers integrate these high-level features and perform classification. In our study, the feature maps from the final convolutional layer of the model captured the high-level features of AD. Therefore, by using Grad-CAM to visualize the feature maps from the last layer, the disease-related features can be visualized, and the ROIs associated with the disease can be identified, which provides informative guidance for disease research and improves the performance of the model.

As shown in Fig 4, the ADNI dataset performed worse on 3D-VGG16 than on 3D-VGG16-GAP, which is likely due to the inappropriate high-level features extracted by the model. As shown in Table 2, the accuracy of the fALFF map was the lowest, corresponding to the fact that the distribution of the highlighted regions was random, and the model did not find any specific ROIs to focus on. By contrast, the accuracy of the VMHC map was also relatively low; however, the model focused on almost the entire brain, as shown in Fig 4. For the ALFF, ReHo, and combined maps, which had higher accuracy, the model tended to highlight a specific ROI. We also believe that the improved imaging performance of Grad-CAM heatmaps on 3D-VGG16-GAP can be attributed to the fact that the GAP layer preserves the spatial structure information to a greater extent, in contrast to the fully connected layers in 3D-VGG16

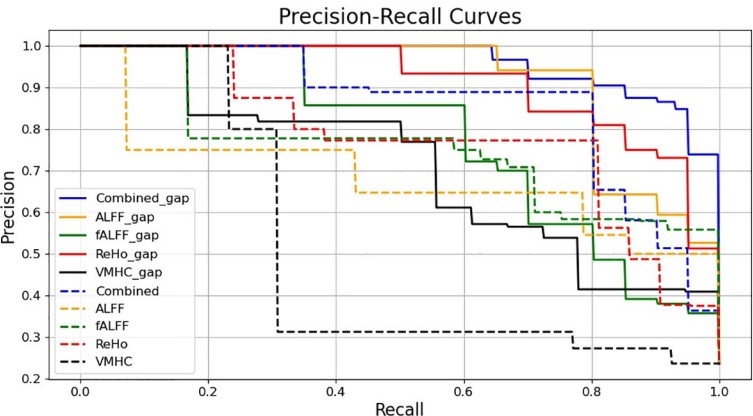

**Fig 8. Precision-Recall curves of 3D-VGG16 and 3D-VGG16-GAP.**

which tend to destroy such information due to the flatten operation. Fig 8 shows the Precision-Recall curves of 3D-VGG16 and 3D-VGG-GAP. For the well-preforming resting-state functional activity maps, such as ALFF, ReHo and combined maps, the performance of 3D-VGG16-GAP consistently surpasses that of its corresponding 3D-VGG16 counterpart, which can be considered as support for the aforementioned discussion.

As shown in Tables 2 and 3, the overall accuracy of 3D-VGG16-GAP was higher than that of 3D-VGG16. Intuitively, the Grad-CAM heatmaps in Fig 5 also show that the effect is better than that shown in Fig 4. For the NC category, 3D-VGG16-GAP focused on the entire brain. Specific regions are highlighted in the AD category for comparison. Compared with that in Fig 6, all the Grad-CAM heatmaps cover the precuneus. In some cases, such as the heatmaps of the ReHo, VMHC, and combined one, the hippocampus is also covered.

The precuneus is located inside the brain between the two cerebral hemispheres in the posterior region between the somatosensory cortex and anterior to the cuneus. It has various cognitive and neural functions, including spatial perception, visual attention, perception and consciousness, memory, self-awareness, and emotion processing. Some studies suggest that atrophy and pathology of the precuneus are the main causes of AD. [57] indicated that patients with early-onset AD exhibit a significant and distinct reduction in precuneus size, which is not observed to the same extent in patients with late-onset AD. In addition, the activity of choline acetyltransferase in the precuneus was found to be significantly lower in individuals with AD than in NC and similar between individuals with mild cognitive impairment and NC [58]. There is novel evidence of a difference in repetitive transcranial magnetic stimulation of the precuneus between patients with AD and NCs [59]. In our study, the highlighted ROIs overlay the precuneus area, suggesting that the model's classification is probably based on this area. Thus, the primary difference between NC and AD is likely located within the precuneus area. This supports and corroborates prior medical research results.

In both primates and humans, the hippocampus is a relatively small structure located in the medial aspect of the temporal lobe, adjacent to the lateral ventricle, and is typically characterized by its horseshoe-shaped morphology, hence its name "hippocampus". According to several studies [60–63], the hippocampus is one of the earliest regions in the brain that experience damage in various forms of dementia, including AD. This is because the hippocampus is responsible for memory formation and retrieval and plays a critical role in the formation of new memories. As dementia progresses, hippocampal damage worsens, leading to memory impairments and other cognitive deficits.

As previously stated, it is believed that for complex datasets such as fMRI, the more manual feature extraction, the less the model needs to do, and thus the better the model's performance may be. However, manually extracted features may not be the features required for classification by the model, indicating that they may be not discriminative enough for the model. Therefore, the required features for different fMRI data of diseases may vary, and the same method is likely to be ineffective for different diseases. In our study, several 3D resting state functional activity maps extracted from 4D fMRI data are utilized as the dataset, which differs from using 2D or 3D slices of 4D fMRI data. This approach eliminates data leakage and resting state functional activity maps possess higher-level features than raw slices. This may alleviate the pressure on the model for feature extraction, thus improving the model's performance. Additionally, using 3D resting-state functional activity maps could also retain more spatial information in the model's feature maps, which could enhance the explainability of the model. Even for AD, due to the small size of the dataset used in this study, which is a common issue in medical datasets due to the strict data collection conditions and the high cost of acquiring equipment, the generalizability is still limited, and further validation is required to determine its applicability to other AD patients.

The lack of explainability in deep learning models has been a longstanding issue; however, it has been partially addressed by the introduction of CAM methods. In our study, Grad-CAM was utilized to explain the model, which not only contributes to the explainability of the model for the diagnosis and research of the disease, but also helps improve the accuracy of the model by studying the cases that were incorrectly predicted. However, because of the inherent limitations of CAM methods [23, 24], in which the feature maps with the maximum resolution of $14 \times 14 \times 14$ are used, even after interpolation, the imaging scope is coarse. In medical imaging, this can undoubtedly affect the accurate localization of lesions.

As mentioned above, the LRP method [32] was proposed to solve the problem of low resolution of CAM methods. However, the LRP method only takes into account model parameters and neuron activations. By this, the heatmaps are less prone to group effects in the data because they are produced individually. The LRP method is very specific for individuals with high inter-patient variability, which is not conducive to the detection of the common features among patients with the same disease. Unlike LRP method, the generation of Grad-CAM heatmaps not only relies on feature maps individually but also takes into account the shared parameters of gradients and other model-related features. The combination of these two points may result in superior performance when extracting common features among different individuals.

In future research, we hope to develop a new method for extracting high-level features without reducing the resolution of feature maps to address the aforementioned limitations. In addition, the improved CAM method could help fix this limitation by explaining models that are unrestricted by the resolutions of feature maps. In addition, considering that the brain is mutually interconnected, with different ROIs forming a topological map based on the strength of their connections with each other, other forms of deep learning models, such as graph neural networks (GNN), may be used to achieve higher accuracy in disease classification [52, 64]. Additionally, relatively little research has been conducted on explainability methods based on GNNs [65], which may be another promising direction for further exploration.

## Conclusion

In this study, we applied Grad-CAM to an fMRI-based 3D-VGG16 network for AD diagnosis to substantiate its validity, thereby achieving the localization of AD-related ROIs. In addition,

the use of resting-state functional activity maps as the dataset successfully reduced the complexity of the fMRI data, facilitating more efficient feature extraction by the model. Grad-CAM helped achieve the precise localization of disease lesions and analysis of the reasons for misclassification. The results showed that during the prediction process following training, the ROIs on which the model focused were almost identical to the areas where lesions have been shown in current research on AD. This supports and corroborates current research and facilitates the use of deep learning to study other diseases with unknown etiology. However, issues with localization accuracy are still present. Improving the deep learning models, changing the types of deep learning models, and improving the CAM method may help alleviate this limitation.

## Acknowledgments

The data utilized in this article were obtained from the Alzheimer's Disease Neuroimaging Initiative (ADNI) database (adni.loni.usc.edu). As such, the investigators within the ADNI contributed to the design and implementation of ADNI and/or provided data but did not participate in analysis or writing of this report. A complete list of ADNI investigators can be found at: http://adni.loni.usc.edu/wp-content/uploads/how_to_apply/ADNI_Acknowledgement_List.pdf. ADNI data are disseminated by the Laboratory for Neuro Imaging at the University of California, Los Angeles. We thank Karen Klein MA, ELS (Translational Science Institute, Wake Forest University Health Sciences), for editing the manuscript. We express our gratitude to the anonymous reviewers for their valuable and insightful feedback.

## Author Contributions

**Data curation:** Boyue Song.

**Formal analysis:** Boyue Song.

**Funding acquisition:** Shinichi Yoshida.

**Investigation:** Boyue Song.

**Methodology:** Boyue Song.

**Resources:** Shinichi Yoshida.

**Software:** Boyue Song.

**Supervision:** Shinichi Yoshida.

**Validation:** Boyue Song.

**Visualization:** Boyue Song.

**Writing – original draft:** Boyue Song.

**Writing – review & editing:** Shinichi Yoshida.

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
