## [Decision Letter · Decision Letter 0]

20 Feb 2024

PONE-D-23-40870Explainability of Three-dimensional Convolutional Neural Networks for Functional Magnetic Resonance Imaging of Alzheimer’s Disease Classification based on Gradient-weighted Class Activation MappingPLOS ONE

Dear Dr. Song,

Thank you for submitting your manuscript to PLOS ONE. After careful consideration, we feel that it does not fully meet PLOS ONE publication criteria as it currently stands. Therefore, we invite you to submit a revised version of the manuscript that addresses the points raised during the review process.

Your manuscript has been carefully reviewed the reviewers and received mixed comments. Please address these mandatory queries (appended below) line-by-line in your Response Letter document with specific details about any changes that were made in your revised manuscript, or the reasons why the suggested changes have not been made.

We look forward to receiving your revised manuscript.

Kind regards,

Arka Bhowmik, Ph.D.

Academic Editor

PLOS ONE

2. PLOS requires an ORCID iD for the corresponding author in Editorial Manager on papers submitted after December 6th, 2016. Please ensure that you have an ORCID iD and that it is validated in Editorial Manager. To do this, go to ‘Update my Information’ (in the upper left-hand corner of the main menu), and click on the Fetch/Validate link next to the ORCID field. This will take you to the ORCID site and allow you to create a new iD or authenticate a pre-existing iD in Editorial Manager. Please see the following video for instructions on linking an ORCID iD to your Editorial Manager account: " ext-link-type="uri" xlink:type="simple">https://www.youtube.com/watch?v=_xcclfuvtxQ".

 [This work was supported by Japan Society for the Promotion of Science KAKENHI, Grant Numbers JP22K12786, JP22K19650, JP21H03553, JP22H03699, and JP20H00267, China Scholarship Council 202106030072].  

[This work was supported by JSPS KAKENHI, Grant Numbers JP22K12786,

JP22K19650, JP21H03553, JP22H03699, and JP20H00267, China Scholarship Council

202106030072. ]

 [This work was supported by Japan Society for the Promotion of Science KAKENHI, Grant Numbers JP22K12786, JP22K19650, JP21H03553, JP22H03699, and JP20H00267, China Scholarship Council 202106030072]. 

5. Please provide a complete Data Availability Statement in the submission form, ensuring you include all necessary access information or a reason for why you are unable to make your data freely accessible. If your research concerns only data provided within your submission, please write "All data are in the manuscript and/or supporting information files" as your Data Availability Statement.

7. We note that figure 1, 2, 4, 5, 6, 7, 8, 9 and 10 in your submission contain copyrighted images. All PLOS content is published under the Creative Commons Attribution License (CC BY 4.0), which means that the manuscript, images, and Supporting Information files will be freely available online, and any third party is permitted to access, download, copy, distribute, and use these materials in any way, even commercially, with proper attribution. For more information, see our copyright guidelines: http://journals.plos.org/plosone/s/licenses-and-copyright.

a. You may seek permission from the original copyright holder of figure 1, 2, 4, 5, 6, 7, 8, 9 and 10 to publish the content specifically under the CC BY 4.0 license. 

Reviewers' comments:

Reviewer's Responses to Questions

**Comments to the Author**

1. Is the manuscript technically sound, and do the data support the conclusions?

Reviewer #1: Yes

Reviewer #2: Partly

2. Has the statistical analysis been performed appropriately and rigorously? 

Reviewer #1: Yes

Reviewer #2: N/A

3. Have the authors made all data underlying the findings in their manuscript fully available?

Reviewer #1: Yes

Reviewer #2: No

4. Is the manuscript presented in an intelligible fashion and written in standard English?

Reviewer #1: Yes

Reviewer #2: Yes

5. Review Comments to the Author

Reviewer #1: General comments

In the introduction section, the author highlights prior studies that employed various deep learning models, often achieving high accuracy. Instead of merely presenting these findings, it is recommended to emphasize the limitations of these studies. Subsequently, you can refer to these limitations when introducing your proposed method.

The study encompasses a substantial amount of information that requires careful consideration. Notably, there is an absence of explanations for terms such as "VGG" and "blood oxygen" within the document. The background section tends to include superfluous words without providing essential details. It is recommended that the author concentrates on the study, highlights the limitations of previous studies, and underscores the advantages of the proposed methodology. Furthermore, it would be advantageous to verify the appropriate terminology for imaging modalities, specifically determining whether to refer to fMRI images, MRI images, or MR images.

A lack of elucidation connects your proposed study and blood oxygenation levels. Additionally, the absence of a detailed explanation of the preprocessing steps in the study contributes to a challenge in understanding. Providing clarity on these aspects would significantly enhance comprehension for readers.

Additionally, the study consistently mentions brain regions without providing a detailed explanation. It's essential to acknowledge that brain regions are intricate and multifaceted, and a mere mention without elaboration may not suffice, particularly in the context of classification tasks. A more in-depth exploration or clarification of the relevance of specific brain regions to the classification task is recommended for a more robust interpretation. It appears that this paper is stronger in technical aspects rather than providing satisfactory detail in medical explanations. It may be beneficial to incorporate more comprehensive medical explanations to enhance the overall quality, ensuring a well-balanced and informative presentation across both technical and medical dimensions.

Specific comments

1. The abstract sessions: "The aim is to explore the specific brain regions the model focuses on…." And "….the highlighted brain regions were primarily located in the precuneus and the hippocampus for AD subjects….."

2. The sentences mentioned above manifest within the context of your model explanation. For enhanced clarity and comprehension, it is recommended that you articulate them initially in your discourse.

3. Line 4: Enhanced complet eness can be achieved by incorporating references into the content.

4. Line 10: These sentences delineate the advantages of utilizing fMRI; however, the rationale for employing fMRI merely emphasizes its "high" attributes. The presented reasons lack substantive depth and may not suffice for inclusion in your manuscript.

5. Line 13: Instead of " In [7], use the author's name.

6. Line 38 and 41: Although your results exhibit a high level of accuracy, they still fall below the benchmarks established by prior studies. It is imperative to identify and elucidate the limitations inherent in these earlier studies compared to your result, particularly in classification tasks.

7. Line 46: In using raw fMRI data in your study, specifying the MRI image view is advisable. Explicitly state the chosen view and elucidate its significance, as the selected MRI image view has the potential to contribute significantly to achieving higher accuracy. Please refer to this paper (Y. Pusparani et al., "Diagnosis of A"Alzheimer'sDisease UsingAlzheimer's Convolutional Neural Network With Select Slices by Landmark on Hippocampus in MRI Images,""in IEEE AccesDs, vol. 11, pp. 61688-61697, 2023, doi: 10.1109/ACCESS.2023.3285115).

8. Line 122: The hypothesis of your study introduces novel terms, including "blood oxygen." However, the abstract and background sections lack an elucidation of the concept of blood oxygen. Although the author mentions "brain region," it is crucial to note that brain regions encompass a wealth of information. The author should specify pertinent details related to the mentioned brain region to prevent potential misunderstandings.

9. Line 153: Including a hyperlink to the ADNI website would benefit researchers. Previous studies have incorporated the ADNI website link in their respective investigations.

10. Line 154: Are the 163 scans referred to as slices? The author indicates that the total number of subjects is 50. Is the count of 163 scans (slices) applicable to each subject, thereby summing up to 8,150 (163 x 50)? Utilizing a table for detailed information on the datasets would enhance clarity and provide a comprehensive overview.

11. Line 155: Do you consider the number of subjects relatively small? To strengthen the evidence, it would be advisable to identify previous studies employing comparable subjects. Additionally, exploring the effectiveness of classification tasks in studies with similar subject counts could contribute valuable insights to assess the suitability of the chosen sample size.

12. Line 162: Could you clarify the term "other preprocessing step"? An elaboration on this specific preprocessing step would enhance understanding and enable a more comprehensive interpretation of the methodology.

13. Line 267: Is the batch size considered small?

14. Line 269: Have any other research papers conducted a similar procedure multiple times, such as 10 repetitions? Including evidence from relevant literature to substantiate and support the chosen methodology would be beneficial.

15. Line 271: Is the 20% allocated for testing derived from the same subjects but different slices?

16. Line 275: It is more appropriate to use "mAP" to refer to Mean Average Precision in the context of performance evaluation, particularly in tasks like object detection or information retrieval. If you are referring to a geographical map or a visual representation, then "map" would be the suitable term.

17. Tables 1 and 2 reveal a consistent decrease in the VMHC results for cross-validation compared to validation, typically by a factor of 2 to 3. It is imperative to delve into a discussion to identify the underlying reasons for this trend, particularly when other instances demonstrate an increase. Analyzing and understanding the factors contributing to the observed variation can enhance the overall interpretation and validity of the results.

18. Table 3 presents various stages of MCI, suggesting using ADNI-2 or ADNI-3 data. However, the background and abstract sections of the paper mention only AD and NC. The author should address and reconcile this discrepancy by clarifying the inclusion of MCI stages in the study, ensuring consistency in the presentation of information.

19. However, this study does not show the reason for using Resnet-18nd 3D-CNN.

20. In Fig. 6, is there any mention by the author regarding the coverage of the hippocampus? If so, it would be beneficial to elaborate on the meaning and implications of this statement for a clearer understanding. The author did not comprehensively explain the hippocampus, a relatively small brain region. It would be advantageous to offer a more detailed and thorough explanation to enhance the reader's understanding of the hippocampus's significance and role in the study's context.

21. Line 335: The study utilizes 3D fMRI images, whereas 4D fMRI images are available. It would be valuable to outline the advantages and disadvantages associated with both 3D and 4D fMRI imaging to provide a comprehensive understanding of the considerations and choices made in the study.

22. Line 339: Are the six patients mentioned the same as the 50 subjects referenced in previous statements? If so, the rationale behind including only six patients needs clarification. It would be beneficial to specify if these six patients represent a subset for each stage or provide additional information on the selection criteria to enhance transparency in the study design.

23. Line 341: What is the indicator of the correct heatmap?

24. Line 342: Could you provide more context or specify the section you are referring to? Without additional information, It is unable to identify the particular region highlighted in the text.

25. Line 346-357: Discussing a specific region requires further consideration. Providing additional details or context about the region in question would contribute to a more comprehensive and focused analysis.

26. The discussion sections: it would be better to compare your result with previous studies since previous studies achieve higher accuracy in the background.

27. Line 417-428: Despite the author's explanation of the precuneus, no correlation was established between the precuneus and AD classification tasks. The explanation seems more geared toward medical reasons than directly addressing its relevance to classification tasks. A more explicit connection or clarification regarding the role of the precuneus in the context of AD classification tasks may be warranted.

28. Line 475: If you characterize this dataset as small, it raises the question of why it was chosen for use in the study. Providing a rationale for selecting a comparatively small dataset would enhance the understanding of the study's design choices and contribute to the overall transparency of the research.

29. Line 483: what are the maximum resolutions?

30. Line 508-509: Indeed, the complexity of brain regions necessitates a more detailed explanation rather than a generic reference to "brain region." Providing specific and comprehensive details about the implicated brain regions would enhance the clarity and depth of the study's neuroscientific context.

31. The authors may refer to the below paper for more deep lerning studies focusing primarily on the hippocampus of AD subjects. (2023). Diagnosis of Alzheimer's Disease Using Convolutional Neural Network With Select Slices by Landmark on Hippocampus in MRI Images. IEEE Access, 11, 61688-61697. doi:10.1109/ACCESS.2023.3285115

Reviewer #2: The authors did a lot of great work to

1. Preprocessing fMRI data in different ways.

2. Train VGG models to classify Alzheimer’s disease.

3. Use Class Activation Mapping methods to explain the VGG models.

The manuscript is well written and easy to follow. The authors did a great job in describing the problem and solutions end-to-end. Also it provided interesting discussion on the relation between model explainability and model prediction performance. However, I think some parts of the study are confusing to me in terms of purposes. In addition, more details need to be provided for some of the conclusions. I listed my questions below.

1. Line 233. The authors chose to add the Global average pooling (GAP) layer to the VGG architecture because they think the GAP layer is required for the Class Activation Mapping method. This claim is valid but the authors didn’t directly use the GAP layer to explain the 3D-VGG16-GAP model, which is the method used by the original CAM paper (Zhou, Bolei, et al. "Learning deep features for discriminative localization." Proceedings of the IEEE conference on computer vision and pattern recognition. 2016.). Instead, all visual explanations in the paper are generated using the Grad-CAM method, which doesn’t require the existence of a GAP layer in the model. I suggest the author provide GAP layer based CAM explanations to better justify adding a GAP layer. Otherwise the GAP layer isn’t necessary because it is not needed by the Grad-CAM methods, which was the only used method for visual explanations in the manuscript.

2. GAP layer advantage. The authors concluded that models with a GAP layer performed better in prediction accuracy because it potentially reduced overfitting (Abstract, Line 240, Line 289, Line 367). First, I think the models with and without the GAP layer only have marginal performance differences. With the small sample size, I suggest providing statistical test results to show that the performance difference is statistically significant. Second, both the original CAM paper (Zhou, Bolei, et al. "Learning deep features for discriminative localization." Proceedings of the IEEE conference on computer vision and pattern recognition. 2016.) and a similar study on 3D MRI images (Yang, Chengliang, Anand Rangarajan, and Sanjay Ranka. "Visual explanations from deep 3D convolutional neural networks for Alzheimer’s disease classification." AMIA annual symposium proceedings. Vol. 2018. American Medical Informatics Association, 2018.) reported a prediction performance loss because of the GAP layer. I think a more in-depth experiment and discussion is needed to understand under what scenario the performance loss is overturned as a benefit.

3. Imaging performance discussion. In multiple statements in the manuscript (Figure 4, Line 410), the authors think the fully connected layers destroyed the 3D feature maps structure so the imaging performance is worse compared to the models with the GAP layer. First, imaging performance wasn’t well defined as a metric in the manuscript. From my understanding, this conclusion is drawn from visual inspections by the authors. But only few visual explanation samples are provided to support this conclusion. To make this claim well supported, I think the authors need to provide quantitative measures from bigger samples of data. A similar study (Yang, Chengliang, Anand Rangarajan, and Sanjay Ranka. "Visual explanations from deep 3D convolutional neural networks for Alzheimer’s disease classification." AMIA annual symposium proceedings. Vol. 2018. American Medical Informatics Association, 2018.) used the voxel level precision-recall curve for identifying cerebral cortex, lateral ventricle, and hippocampus regions as the quantitative measure. The original Grad-CAM paper also employed quantitative measures (Selvaraju, Ramprasaath R., et al. "Grad-cam: Visual explanations from deep networks via gradient-based localization." Proceedings of the IEEE international conference on computer vision. 2017.).

4. Train-test split. In line 325, the authors described the experiment's performances and explanations with raw fMRI data. Later in the discussion section (line 446), the authors pointed out that there was a label information leakage between train and test split in the experiment's setup. I think if the authors want to show explanations that can help identify experiment issues, they should move the two experiments (train test split with and without label information leakage) to the experiments section and explicitly state the purpose of using explanation to diagnose models. Currently it gives the reader a surprise when I noticed the entire experiment setup on the raw fMRI data was invalid because of the label information leakage. On the same topic, the author mentioned that “the dataset for our study consisted of 163 scans of fMRI data from 50 healthy controls and 105 scans of fMRI data from 34 patients with AD” (line 154). Therefore, label information leakage could also exist in the preprocessed fMRI experiments if the train-test split is at the scan level. The author mentioned the train-test split was a random 80/20% split but didn’t mention if it is scan or patient level. It would be great if the author could clarify this.

Minor issues.

1. Line 266. The author combined the feature maps of ALFF, fALFF, ReHo, and VMHC. But how the combination is done wasn’t clear.

2. Line 246. The author mentioned the last convolution layer should be used for visual explanation. But I think the Grad-CAM visual explanation can be applied to any convolution layer. I understand the last conv layer is more tied to importance. But I think it is good to call out that Grad-CAM isn’t restricted to the last convolution layer.

3. Fig 1. I think the layer dimension size of the Framework of 3D-VGG16 and 3D-VGG16-GAP part is too blurry to read. I can’t read the exact numbers of the dimension sizes.

4. I can't find the Github link for experiment code and data.

6. PLOS authors have the option to publish the peer review history of their article (what does this mean?). If published, this will include your full peer review and any attached files.

Reviewer #1: **Yes: **Chi-Wen Lung

Reviewer #2: No

---

## [Author Response · Author response to Decision Letter 0]

5 Apr 2024

Dear Editor and Reviewers,

We would like to express our sincere gratitude for your time and valuable feedback on our manuscript titled “Explainability of Three-dimensional Convolutional Neural Networks for Functional Magnetic Resonance Imaging of Alzheimer’s Disease Classification based on Gradient-weighted Class Activation Mapping”. We appreciate the opportunity to revise and provide the necessary clarifications and revisions.

Response to the Journal requirements from Editor:

Response: Thank you for your feedback. We have carefully reviewed the additional requirements for our manuscript and made necessary revisions to address them. Specifically, we have ensured that the formatting of the paper and the file naming meet PLOS ONE’s style guidelines. We appreciate your guidance in this matter.

2. PLOS requires an ORCID iD for the corresponding author in Editorial Manager on papers submitted after December 6th, 2016. Please ensure that you have an ORCID iD and that it is validated in Editorial Manager. To do this, go to ‘Update my Information’ (in the upper left-hand corner of the main menu), and click on the Fetch/Validate link next to the ORCID field. This will take you to the ORCID site and allow you to create a new iD or authenticate a pre-existing iD in Editorial Manager. Please see the following video for instructions on linking an ORCID iD to your Editorial Manager account: https://www.youtube.com/watch?v=_xcclfuvtxQ". 

Response: We have obtained and validated the ORCID iD for the corresponding author in Editorial Manager as per the requirement. Thank you for your guidance in ensuring compliance with PLOS ONE’s policy regarding ORCID iD.

 [This work was supported by Japan Society for the Promotion of Science KAKENHI, Grant Numbers JP22K12786, JP22K19650, JP21H03553, JP22H03699, and JP20H00267, China Scholarship Council 202106030072]. 

Response: Thank you for acknowledging the financial disclosure and providing us with the opportunity to clarify the role of the funders in our study. We would like to provide our financial disclosure as following:

This work was supported by Japan Society for the Promotion of Science KAKENHI, Grant Numbers JP22K12786, JP22K19650, JP21H03553, JP22H03699, and JP20H00267, China Scholarship Council 202106030072. The funders had no role in study design, data collection and analysis, decision to publish, or preparation of the manuscript.

[This work was supported by JSPS KAKENHI, Grant Numbers JP22K12786,

JP22K19650, JP21H03553, JP22H03699, and JP20H00267, China Scholarship Council

202106030072. ]

 [This work was supported by Japan Society for the Promotion of Science KAKENHI, Grant Numbers JP22K12786, JP22K19650, JP21H03553, JP22H03699, and JP20H00267, China Scholarship Council 202106030072]. 

Response: Thank you for reminding us about the funding information. We would like to provide the following elucidation regarding our funding information: 

This work was supported by Japan Society for the Promotion of Science KAKENHI, Grant Numbers JP22K12786, JP22K19650, JP21H03553, JP22H03699, and JP20H00267. The recipient of the above fundings is Shinichi Yoshida. This work was also supported by China Scholarship Council, Award Number 202106030072. The recipient is Boyue Song. 

5. Please provide a complete Data Availability Statement in the submission form, ensuring you include all necessary access information or a reason for why you are unable to make your data freely accessible. If your research concerns only data provided within your submission, please write "All data are in the manuscript and/or supporting information files" as your Data Availability Statement.

Response: Thank you for reminding us about the Data Availability Statement. We would like to provide our Data Availability Statement as follows:

The authors do not own data used in the manuscript. Data obtained were collected and owned by the Alzheimer’s Disease Neuroimaging Initiative. Researchers may request and access the data through the website of the Alzheimer’s Disease Neuroimaging Initiative (ADNI) (http://adni.loni.usc.edu). Authors had no special access privileges to this data. 

Response: Thank you for reminding us about the ethics statement. Following your advice, we have moved the ethics statement to the Methods section in line 163. The ethics statement is stated as follows:

Data used in the preparation of this article were obtained from the Alzheimer’s Disease Neuroimaging Initiative (ADNI) database (adni.loni.usc.edu). The ADNI was launched in 2003 as a public-private partnership, led by Principal Investigator Michael W. Weiner, MD. The primary goal of ADNI has been to test whether serial magnetic resonance imaging (MRI), positron emission tomography (PET), other biological markers, and clinical and neuropsychological assessment can be combined to measure the progression of mild cognitive impairment (MCI) and early Alzheimer’s disease (AD). All ADNI studies are conducted according to the Good Clinical Practice guidelines, the Declaration of Helsinki, and U.S. 21 CFR Part 50 (Protection of Human Subjects) and Part 56 (Institutional Review Boards). Written informed consent was obtained from all participants before protocol-specific procedures were performed. The Institutional Review Boards approved the ADNI protocol of all participating institutions; for up-to-date information, see www.adni-info.org.

7. We note that figure 1, 2, 4, 5, 6, 7, 8, 9 and 10 in your submission contain copyrighted images. All PLOS content is published under the Creative Commons Attribution License (CC BY 4.0), which means that the manuscript, images, and Supporting Information files will be freely available online, and any third party is permitted to access, download, copy, distribute, and use these materials in any way, even commercially, with proper attribution. For more information, see our copyright guidelines: http://journals.plos.org/plosone/s/licenses-and-copyright. 

Response: Thank you for reminding us about the copyrighted images. The raw fMRI images in Figure 1 are sourced from the ADNI dataset. Due to copyright restrictions, we have removed them from Figure 1. Besides, the other MRI images used as background in Figures 1, 2, 4, 5, and 7 are sourced from MNI152_T1_2mm_brain.nii template within DPABI software, which does not raise copyright concerns. Relevant information regarding DPABI has been appropriately cited in the paper. Due to copyright considerations, we have also replaced the background in Figure 6 with the MNI152 template. Finally, after considering the valuable feedback from the reviewers and engaging in thorough discussions, we have decided to remove the original Figures 8, 9, 10, along with corresponding textual sections.

Response to the comments from Reviewer 1:

Thank you very much for providing valuable feedback. Below is our point-by-point response to your comments. 

1. The abstract sessions: "The aim is to explore the specific brain regions the model focuses on…." And "….the highlighted brain regions were primarily located in the precuneus and the hippocampus for AD subjects….."

2. The sentences mentioned above manifest within the context of your model explanation. For enhanced clarity and comprehension, it is recommended that you articulate them initially in your discourse. 

Response: Thank you for your suggestions regarding the narrative logic. We have revised above sentences as follows to enhance clarity and comprehension:

The aim is to explore the specific Region of Interest (ROI) of brain the model primarily focuses on when making predictions, as well as whether there are differences in these ROIs between AD and normal controls (NCs).

3. Line 4: Enhanced completeness can be achieved by incorporating references into the content.

Response: Thank you for your valuable suggestion regarding the reference citation. We have cited the following reference at location to enhance completeness in line 5:

Donahue, Manus J., et al. "Time delay processing of hypercapnic fMRI allows quantitative parameterization of cerebrovascular reactivity and blood flow delays." Journal of Cerebral Blood Flow Metabolism 36.10 (2016): 1767-1779.

4. Line 10: These sentences delineate the advantages of utilizing fMRI; however, the rationale for employing fMRI merely emphasizes its "high" attributes. The presented reasons lack substantive depth and may not suffice for inclusion in your manuscript.

Response: Thank you for your advice on the coherence of the sentences. We intended to highlight the limitations of using raw fMRI data due to its high complexity, to underscore its impracticality and introduce the resting-state functional activity map method used in our study. We apologize for any confusion caused by our unclear explanation. To enhance clarity, we have revised the sentence as follows in line 9:

Moreover, due to the relatively high temporal and spatial resolutions of fMRI, the complexity of its data is extremely high, which is another primary reason why fMRI data is hard to be directly utilized for medical analysis and diagnosis.

5. Line 13: Instead of “In [7]”, use the author's name.

Response: Thank you for your valuable advice regarding the proper citation of references. We have revised the original sentence according to your advice in line 15:

Tang et al. proposed a method in which each brain was registered to MNI standard space and subdivided into 90 regions, and the regional time series were obtained by calculating the average fMRI signal across all voxels in each region.

6. Line 38 and 41: Although your results exhibit a high level of accuracy, they still fall below the benchmarks established by prior studies. It is imperative to identify and elucidate the limitations inherent in these earlier studies compared to your result, particularly in classification tasks.

Response: Thank you for your feedback regarding the narrative. We concluded this paragraph by adding the following sentence to elucidate the limitations inherent in prior studies in line 54:

Although these methods demonstrate high accuracy, their models lack explainability, thereby diminishing the credibility of the model due to the inability to explain its predictions.

7. Line 46: In using raw fMRI data in your study, specifying the MRI image view is advisable. Explicitly state the chosen view and elucidate its significance, as the selected MRI image view has the potential to contribute significantly to achieving higher accuracy. Please refer to this paper (Y. Pusparani et al., "Diagnosis of A"Alzheimer'sDisease UsingAlzheimer's Convolutional Neural Network With Select Slices by Landmark on Hippocampus in MRI Images,""in IEEE Access, vol. 11, pp. 61688-61697, 2023, doi: 10.1109/ACCESS.2023.3285115).

Response: Thank you for your advice regarding the narrative completeness. We have supplemented the information regarding the MRI image view of prior study in line 41:

The fMRI data were preprocessed using a standard pipeline, sliced into two-dimensional images from axial view and time axes, and randomly divided into training and testing datasets.

Besides, after considering the valuable feedback from the reviewers and engaging in thorough discussions, we have decided to remove our method, experiment and discussion related to deep learning method based on raw fMRI data.

8. Line 122: The hypothesis of your study introduces novel terms, including "blood oxygen." However, the abstract and background sections lack an elucidation of the concept of blood oxygen. Although the author mentions "brain region," it is crucial to note that brain regions encompass a wealth of information. The author should specify pertinent details related to the mentioned brain region to prevent potential misunderstandings.

Response: Thank you for your correction regarding our unclear concept description. We have provided an explanation of the concept of “blood oxygen” in line 133 as follows:

Blood oxygenation level refers to the proportion of oxygen bound hemoglobin, which can be used to infer brain activity, as neural activity induces alterations in local blood flow and oxygenation levels. 

Additionally, we apologize for the confusion caused by our unclear description of “brain region”. We have replaced all instances of “brain region” with the medical term “ROI” (Region of Interest), which refers to a specific area or region within an image or dataset that is selected for further analysis or investigation due to its relevance to a particular research question or clinical condition.

9. Line 153: Including a hyperlink to the ADNI website would benefit researchers. Previous studies have incorporated the ADNI website link in their respective investigations.

Response: Thank you for your valuable advice. We have included a hyperlink to the ADNI website in line 176.

10. Line 154: Are the 163 scans referred to as slices? The author indicates that the total number of subjects is 50. Is the count of 163 scans (slices) applicable to each subject, thereby summing up to 8,150 (163 x 50)? Utilizing a table for detailed information on the datasets would enhance clarity and provide a comprehensive overview.

Response: Thank you for your questions and suggestions. We apologize for any confusion caused by our unclear description. In the ADNI dataset, some subjects underwent multiple scans, resulting in multiple sets of scan data. For AD, a total of 163 scan data were obtained from 50 subjects, while for NC, 105 scan data were obtained from 34 subjects. Additionally, to prevent data leakage, we performed random subject-level partitioning when dividing the dataset. Besides, we have added Table 3 in the manuscript to present detailed information about the ADNI dataset used in our study. We have added an explanation regarding this in line 176:

Based on multiple scans obtained at various time points for each subject, the dataset for our study consisted of 163 scans of fMRI data from 50 NCs and 105 scans of fMRI data from 34 patients with AD, which implies that some subjects possess multiple sets of scan data.

11. Line 155: Do you consider the number of subjects relatively small? To strengthen the evidence, it would be advisable to identify previous studies employi

---

## [Decision Letter · Decision Letter 1]

23 Apr 2024

Explainability of Three-dimensional Convolutional Neural Networks for Functional Magnetic Resonance Imaging of Alzheimer’s Disease Classification based on Gradient-weighted Class Activation Mapping

PONE-D-23-40870R1

Dear Dr. Song,

We’re pleased to inform you that your manuscript has been judged scientifically suitable for publication and will be formally accepted for publication once it meets all outstanding technical requirements.

Kind regards,

Arka Bhowmik, Ph.D.

Academic Editor

PLOS ONE

Reviewers' comments:

Reviewer's Responses to Questions

**Comments to the Author**

1. If the authors have adequately addressed your comments raised in a previous round of review and you feel that this manuscript is now acceptable for publication, you may indicate that here to bypass the “Comments to the Author” section, enter your conflict of interest statement in the “Confidential to Editor” section, and submit your "Accept" recommendation.

Reviewer #1: All comments have been addressed

Reviewer #2: All comments have been addressed

2. Is the manuscript technically sound, and do the data support the conclusions?

Reviewer #1: Yes

Reviewer #2: Yes

3. Has the statistical analysis been performed appropriately and rigorously? 

Reviewer #1: Yes

Reviewer #2: Yes

4. Have the authors made all data underlying the findings in their manuscript fully available?

Reviewer #1: Yes

Reviewer #2: (No Response)

5. Is the manuscript presented in an intelligible fashion and written in standard English?

Reviewer #1: Yes

Reviewer #2: (No Response)

6. Review Comments to the Author

Reviewer #1: Presented in a detailed and thorough manner, the authors have addressed my major concerns and addressed the comments from the previous review.

Reviewer #2: Not sure if it is due to PDF compression, the revised figure 1 is still kind of blurry. It would be great if the author can make it in higher resolution.

Thanks for addressing all comments.

7. PLOS authors have the option to publish the peer review history of their article (what does this mean?). If published, this will include your full peer review and any attached files.

Reviewer #1: **Yes: **Chi-Wen Lung

Reviewer #2: No

---

## [Editor Report · Acceptance letter]

30 Apr 2024

PONE-D-23-40870R1 

PLOS ONE

Dear Dr. Song, 

I'm pleased to inform you that your manuscript has been deemed suitable for publication in PLOS ONE. Congratulations! Your manuscript is now being handed over to our production team.

Kind regards, 

on behalf of

Dr. Arka Bhowmik 

Academic Editor

PLOS ONE